# Derivation of Flow Rate and Calibration Method for High-Volume Air Samplers

Richard Hann[1], Mark Hermanson[2]

[1]Department of Cybernetics, Norwegian University of Science and Technology (NTNU), Trondheim, NO 7491, Norway
[2]Hermanson & Associates LLC, Minneapolis, MN 55419, USA

*Correspondence to*: Richard Hann (richard.hann@ntnu.no)

**Abstract.** Sampling the atmosphere to analyse contaminants is different from other environmental matrices because measuring the volume of air collected requires a mechanical flow-through device to draw the air and measure its flow rate. The device used must have the capability of concentrating the analytes of interest onto a different substrate because the

volumes of air needed are often in the hundreds of cubic meters. The use of high-volume air samplers has grown since 1967 when recommended limits of a large number of organic contaminants in air were developed. Equations used for calculating the air flow through the device over time have similarly been developed. However, the complete derivation of those equations has never appeared in the scientific literature. Here a thorough derivation of those equations is provided with definitions of the mechanical systems that are used in the process, along with the method of calibrating and calculating air

flow.

## 1 Introduction

Collecting environmental samples of the atmosphere is inherently different from sampling soil, ice, snow, water, or organic matter. With the non-atmospheric matrices, the chemical analytes of interest are specific to a typically small and easily measured volume or mass. The atmosphere is a matrix with a significantly lower density which raises the question of how to

collect and measure the large volume of air where the analytes are found.

The first mention in the scientific literature of high-volume (Hi-Vol) air flow regulation was from a toxicological study by Drinker et al. (1937). In this study, the amount of chlorinated biphenyl released to specific amounts of air had to be known to identify the amount of toxic substance inhaled by the test organism. The air flow was measured by an orifice calibrator that enabled the volume of air to be known over a certain time period. Although the orifice calibrator is mentioned in this report,

the calibration system, including the system of equations used for calculating flow, is not identified.

Following the passage of the U. S. Clean Air Act in 1963, a group of U. S. health experts formed a group known as the American Conference of Governmental Industrial Hygienists (ACGIH). In 1966, ACGIH developed recommended no-toxic effect concentration limits in air of 78 different contaminants, many of them organic compounds (Danielson, 1967). The

development of this list led to a requirement to make an air sampler capable of handling large volumes of air because the toxic amounts on the ACGIH list were very low and would be found in low concentrations. Development of Hi-Vol samplers began soon after with early designs using vacuum systems that generated large flow rates (Jutze & Foster, 1967). After further development, these systems were found to provide reproducible results (Clements et al., 1972) and eventually to be reliable in severe weather (Salamova et al. 2014) and robust over many years when properly maintained (Salamova et al., 2016).

The early development of vacuum-assisted Hi-Vol samplers required a system for measuring the volume of air flow through the sampler. While Hi-Vol manufacturers and the literature now provide equations used for this process, none of them includes any derivation of those calculations or discussion about why the variables in the equations are used. The situation is typical of a textbook by Wight (1994) where the basic fluid dynamic principles required for the calculations are outlined, but ultimately the equations are not derived comprehensively. Similarly, the coursework provided by the Air Pollution Training Institute on air sampling (APTI, 1980) presents only the calibration equations along with a multitude of numerical examples, without explaining their origin. Even governmental regulations (40 CFR Appendix B Part 50, US EPA, 2011) and guidelines (US EPA, 1999) focus on the calibration of Hi-Vol samplers but do not derive the procedure in detail. The early literature does not elaborate on the calibration equations. For example, Lynam et al. (1969) investigate different calibration methods for Hi-Vol samplers, showing that significant differences can occur. Similarly, Lee et al. (1972) investigate different methods for measuring suspended particles in air and elaborate in detail on the calibration process of Hi-Vol samplers without deriving any equations. As recently as 2013, ASTM International (2013), in Method D6209-13 for collection of Hi-Vol samples, leaves several blanks in sections covering flow control, flow calibration, calibration orifice, and rootsmeter (sections 9.1.2, 9.1.3, 9.1.4 and 9.1.5), all of which are critical to proper calibration. In the calibration section of this method (12.1), there are references to these blanks in section 9.1. Most studies of atmospheric contaminants collected with Hi-Vol samplers assume that the calibration procedure is understood and rarely discuss calibration details and never include the equations used, including Hermanson & Hites, 1989, Monosmith & Hermanson, 1996, Hermanson et al., 1997, Hermanson et al., 2003, Hermanson et al., 2007, Basu et al., 2009, Salamova et al., 2014, and Hites, 2018.

The objective here is to derive the calculations required for measurement of air flow, volume, and calibration of a Hi-Vol air sampler that are missing from the scientific literature. These calculations are based on principles of fluid dynamics. The results developed provide the air sampling community with the missing derivation of equations that are based on the physical features of a Hi-Vol system. The outcome will improve an air pollution investigator's understanding of the operational features of Hi-Vol samplers. Some specialty Hi-Vol samplers, including those for $PM_{10}$, $PM_{2.5}$, and total suspended particulates (TSP) have different flow devices (e.g. particle pre-separators) or different metering systems, so the equations derived and conditions discussed here may not fully apply to them.

## 2 Measuring Concentration Flow Rate

The following presents an educational approach explaining the general physical equations required to derive the concentration of airborne particulate and gas phase contaminants (e.g. pesticides, polychlorinated biphenyls, polychlorinated dibenzo-*p*-dioxins and furans, polycyclic aromatic hydrocarbons, flame retardants), with Hi-Vol air samplers. Figure 1 shows a typical device with its main components. An inlet is shielding the internals from the environment. Particles in the air are captured by a filter, which is permeable for the airflow but will retain particulate matter above a threshold size (depending on the filter type). Gas phase contaminants are captured with tubes of polyurethane foam (PUF) or other adsorbent substrates (e.g. resin). A flow meter, such as a venturi nozzle with an attached differential pressure gauge, is required to determine the air flow velocity inside the device. The necessary vacuum to force air through the sampler is provided by a pump. A timer connected to the pump measures elapsed sampling time. The air flow rate can be adjusted with a valve. The air that has passed through these filters vents back to the atmosphere via an outlet exhaust pipe. The objective of this sampling is to determine a concentration $C$ of a mass of contaminants $m$ in a sampled volume of air $V$.

$$C = \frac{m}{V} \tag{1}$$

The mass of captured particles can be obtained by weighing the filter before and after the sampling. The weight difference $\Delta m$ will be equal to the mass of captured particles. There is a large sensitivity of the concentration results to errors in weighing, hence special care is advised when handling the filters. When the mass of particles is known, they can be processed further to determine the mass of each contaminant, by using various analytical techniques, e.g. those used for various flame retardants by Salamova et al. (2014). If contaminants in the gas phase are investigated, such as pesticides (Hermanson et al., 2007), additional analytical methods must be applied.

The second physical variable required is the volume of the sampled air $V$. This volume cannot be measured directly. Instead, it is derived by determining the volume of air passing through the sampler per unit time (volume flow rate $\dot{V}$), multiplied with the sampling duration $t$.

$$C = \frac{m}{\dot{V} \cdot t} \tag{2}$$

The elapsed sampling time is quantified by using the timer clock mentioned above. The flow rate is determined using the continuity equation: assuming steady flow conditions, the flow rate can be calculated with the flow velocity $v$ through a given flow cross section $A$.

$$\dot{V} = A \cdot v \tag{3}$$

The flow velocity is measured with a flow device such as a venturi nozzle or an orifice plate shown in Fig. 2 and 3. These flow devices exhibit a specific geometry with a given inlet cross-section ① and a constriction ② shown in Fig. 2. The areas of the cross-sections $A_1$ and $A_2$ are known. By assuming continuity (no leaks), the flow rates through each cross-section must be identical $\dot{V}_1 = \dot{V}_2$. Next, Bernoulli's principle of energy conservation is applied to derive the flow velocities from this system. Bernoulli is stating that for incompressible flow (such as in this example) the energy along a streamline is constant. The energy occurs in three different forms: as static pressure $p$, as dynamic pressure $\frac{\rho}{2}v^2$ and as hydrostatic pressure $\rho g h$, with the fluid density $\rho$, the standard gravity $g$ and the hydrostatic height $h$.

$$p_1 + \frac{\rho}{2}v_1^2 = p_2 + \frac{\rho}{2}v_2^2 \quad \text{and} \quad A_1 \cdot v_1 = A_2 \cdot v_2 \tag{5}$$

$$v_1 = \sqrt{\frac{\Delta p}{c_1 \cdot \rho}} \quad \text{with} \quad \Delta p = p_2 - p_1 \quad \text{and} \quad c_1 = \frac{1}{2} \cdot \left(1 - \frac{A_1^2}{A_2^2}\right) = \text{constant} \tag{6}$$

Note that the hydrostatic pressure is omitted in this case because of the low density of air and a negligible hydrostatic height difference. The flow velocity $v_1$ can be expressed by substituting $v_2$ from the continuity equation into Bernoulli's equation. From this, the flow velocity is derived as a function of pressure difference between the two cross-sections, density, and a constant dimensionless factor $c_1$. The value of this factor can be quantified if the geometry of the flow device is known. However, as it will be shown below, it is not necessary to determine a numerical value. This is applicable for all constant factors that will be introduced throughout the following discussion.

To quantify the velocity $v_1$ – which in turn will be used to calculate the volume flow rate $\dot{V}$ and eventually the concentration $C$ – two new variables must be determined. The differential pressure $\Delta p$ can be measured easily with manometers, ranging from digital instruments to simpler devices such as u-tube manometers. The air density $\rho$ cannot be observed directly and is derived using the ideal gas law, defined by ambient temperature $T_\infty$, ambient pressure $p_\infty$ and the specific gas constant for air $R$.

$$\rho = \frac{p_\infty}{R \cdot T_\infty} \tag{7}$$

Ambient temperature is directly measured with a thermometer and ambient pressure with a barometer. Substituting density with the ideal gas law, Eq. (3) and (6) can be summarized to the following.

$$v_1 = \sqrt{\frac{\Delta p \cdot T_\infty}{c_2 \cdot p_\infty}} \quad \text{with} \quad c_2 = \frac{1}{R} \cdot c_1 = \text{constant} \tag{8}$$

$$\dot{V} = \sqrt{\frac{\Delta p \cdot T_\infty}{c_3 \cdot p_\infty}} \quad \text{with} \quad c_3 = \frac{1}{A_1^2} \cdot c_2 = \text{constant} \tag{9}$$

Note that the constants, $c_2$, $c_3$, are not dimensionless anymore. Equation (9) shows that the volume flow rate is dependent only on the ambient conditions and a pressure difference. Changes in temperature and pressure (i.e. air density) will affect the value of the sampled air volume. This is an unfavorable characteristic for Hi-Vol sampling because it implies that concentration results must be reported along with the ambient conditions during sampling. To allow for easier comparison between measurements, a standardized volume flow $\dot{V}_0$ is introduced. The ambient-condition-specific volume flow rate $\dot{V}$ can be converted to a standardized volume flow by applying the ideal gas law and the standard ambient conditions for temperature ($T_0 = 298.15$ K) and pressure ($p_0 = 1013.25$ hPa).

$$\dot{V}_0 = \dot{V} \cdot \frac{\rho}{\rho_0} = \dot{V} \cdot \frac{p_\infty}{T_\infty} \cdot \frac{T_0}{p_0} = \sqrt{\frac{\Delta p \cdot T_\infty}{c_3 \cdot p_\infty}} \cdot \frac{p_\infty}{T_\infty} \cdot \frac{T_0}{p_0} = \sqrt{\frac{\Delta p \cdot p_\infty \cdot T_0}{c_4 \cdot p_0 \cdot T_\infty}}$$

$$\text{with} \quad c_4 = \sqrt{\frac{p_0}{T_0}} \cdot c_3 = \text{constant}$$

(10)

To underline that this equation is stating standardized volume flow, the pressure and temperature variables are presented as normalized, dimensionless terms, i.e. $\frac{p_\infty}{p_0}$ and $\frac{T_0}{T_\infty}$. Finally, we can include all the above derivations into Eq. (1).

$$C = \frac{m}{V} = \frac{m}{\sqrt{\dfrac{\Delta p \cdot p_\infty \cdot T_0}{c_4 \cdot p_0 \cdot T_\infty}} \cdot t} = f(\Delta m, t, \Delta p, T_\infty, p_\infty)$$

(11)

Equation (11) presents all variables required to be physically measured, necessary to derive the contaminant concentration: contaminant mass, sampling time, differential pressure at the flow device, ambient temperature, and ambient pressure.

## 3 Calibration Method

The necessity to calibrate the volume flow rate arises from the fact that Eq. (11) contains the unknown constant $c_4$. This constant represents not only the constant physical parameters but can also be used to account for second-order effects that have not been included in the equations, such as internal pressure loss, imperfect flow conditions and flow obstructions. Assuming a direct proportional impact of these missing effects, a linear correlation can account for them and also the constant physical parameters. To define this linear correlation, a slope and an intercept must be found. This can be achieved, by using a temporary calibration device to quantify the true, exact flow rate through the system at several pump pressures and to correlate that with Eq. (10). The linear correlation between the true flow rate $\dot{V}_{\text{True}}$ with the unknown flow rate $\dot{V}_0$ can be expressed by introducing a calibration slope ($a_{\text{Calibration}}$) and calibration intercept ($b_{\text{Calibration}}$).

$$\dot{V}_{\text{True}} = \frac{1}{a_{\text{Calibration}}}\left(\dot{V}_0 - b_{\text{Calibration}}\right) = \frac{1}{a_{\text{Calibration}}}\left(\sqrt{\Delta p \cdot \frac{p_\infty \cdot T_0}{p_0 \cdot T_\infty}} - b_{\text{Calibration}}\right) \tag{12}$$

The aim of the calibration process is to determine the numeric value of the calibration slope and intercept. First, the true flow through the air sampler is determined by using a temporary calibration device, typically with an orifice plate (Fig. 3). The true flow is evaluated at several flow rates (adjusted by regulating the pump voltage or the flow valve in Fig.1). Second, the true flow rates are correlated to the differential pressure readings with the aforementioned linear approach in Eq. (12). The method is visualized in Fig. 4.

The calibration process will be described for the example of a *Tisch Environmental Inc. TE-PUF Poly-Urethane Foam High Volume Air Sampler* (Tisch, 2015). This sampling unit is using a venturi nozzle as a flow device and a Magnehelic® differential pressure gage. For the calibration, an *orifice calibrator* is mounted on the sampler. The calibrator consists essentially of a cylindrical can with an orifice plate and a pressure tap (Fig. 3). Despite its simple construction, it is a highly accurate and robust calibration device (Wight, 1994).

To obtain the flow rate through the orifice calibrator $\dot{V}_{\text{Orifice}}$, the same principles (continuity and Bernoulli between ① and ② in Fig. 3) are applied again. Following Eq. (3)–(10), the orifice flow rate will depend on a pressure difference $\Delta p$ between those two reference points. Instead of using a differential pressure gauge, this pressure difference is determined by using a u-tube manometer (slack tube). Bernoulli's principle (between ③ and ④ in Fig. 3) will be used to obtain this pressure difference from the u-tube manometer. One end of the manometer ③ is attached to the pressure tap on the calibration device ($p_3 = p_2$) while the other end ④ is opened to ambient conditions ($p_4 = p_\infty = p_1$).

$$p_3 + \rho_{\text{H2O}} \cdot g \cdot h_3 = p_4 + \rho_{\text{H2O}} \cdot g \cdot h_4 \tag{13}$$

$$\Delta p_{\text{H2O}} = \rho_{\text{H2O}} \cdot g \cdot \Delta h_{\text{H2O}} \quad \text{with} \quad \Delta h_{\text{H2O}} = h_4 - h_3 \tag{14}$$

$$\dot{V}_{\text{Orifice}} = \sqrt{\frac{\Delta h_{H2O} \cdot p_\infty \cdot T_0}{c_5 \cdot p_0 \cdot T_\infty}} \quad \text{with} \quad c_5 = \frac{1}{\rho_{\text{H2O}} \cdot g} \cdot c_4 = \text{constant} \tag{15}$$

Note, the slack tube is filled with water ($\rho_{\text{water}} \approx 1000\,\text{kg/m}^3$), hence the hydrostatic pressure term in the Bernoulli equation cannot be neglected anymore. Because the water in the u-tube is static (flow velocities are zero), the dynamic pressure term vanishes. Equation (15) for the orifice volume flow rate is very similar to Eq. (10). It contains an unknown constant $c_5$ of physical parameters. To determine this constant, and to account for second-order effects, the same principle as for Eq. (12) is applied: the flow rate is correlated with a linear function.

$$\dot{V}_{\text{TrueOrifice}} = \frac{1}{a_{\text{Orifice}}}\left(\dot{V}_{\text{Orifice}} - b_{\text{Orifice}}\right) = \frac{1}{a_{\text{Orifice}}}\left(\sqrt{\Delta h_{H2O} \cdot \frac{p_\infty \cdot T_0}{p_0 \cdot T_\infty}} - b_{\text{Orifice}}\right) \tag{16}$$

The slope $a_{\text{Orifice}}$ and offset $b_{\text{Orifice}}$ are determined in a calibrated, laboratory environment, typically by the manufacturer of the orifice calibrator and provided as documentation for the orifice calibrator. Note that the orifice calibrator needs to be calibrated regularly by the manufacturer in order to maintain the calibration chain (laboratory – calibration device – sampler).

With the pressure difference from the u-tube manometer and the orifice slope and offset, $\dot{V}_{\text{TrueOrifice}}$ can be calculated and the values correlated to the unknown device flow rate $\dot{V}_0$ to obtain the true flow through the sampler $\dot{V}_{\text{True}}$, Eq. (12). For this, several observations $n$ of the flow rate through the calibrator device $\dot{V}_{\text{TrueOrifice}}$ and the sampler flow rate $\dot{V}_0$ are taken. For each observation, readings of the differential pressure gage (Magnehelic®) and the slack tube are taken.

The slope and intercept can be graphically determined by using a linear trend line, by plotting the results of the calibration measurements in a graph shown in Fig. 4. The x-axis represents the flow rate for the orifice calibrator $\left(x = \dot{V}_{\text{TrueOrifice}}\right)$ and the y-axis the flow term for the internal flow device $\left(y = \sqrt{\Delta p \cdot \frac{p_\infty \cdot T_0}{p_0 \cdot T_\infty}}\right)$. The slope and the intercept of the resulting trend line are the sought-after calibration factors $a_{\text{Calibration}}$ and $b_{\text{Calibration}}$ in Eq. (12). Alternative to the graphic solution, the following equations can be applied to numerically determine the slope and intercept.

$$a_{\text{Calibration}} = \frac{\sum(\dot{V}_{\text{TrueOrifice}} \cdot \dot{V}_{\text{True}}) - \frac{\sum \dot{V}_{\text{TrueOrifice}} \cdot \sum \dot{V}_{\text{True}}}{n}}{\sum(\dot{V}_{\text{TrueOrifice}})^2 - \frac{(\sum \dot{V}_{\text{TrueOrifice}})^2}{n}} \tag{17}$$

$$b_{\text{Calibration}} = \frac{\sum \dot{V}_{\text{True}}}{n} - m_{\text{Calibration}} \cdot \frac{\sum \dot{V}_{\text{Orifice\_}}}{n} \tag{18}$$

The results are expected to show a very strong correlation because the flow through the orifice calibrator and the sampling device should be identical. A very low coefficient of correlation, e.g. $r < 0.990$ (Tisch, 2015), could be an indication that there is an error in the system, such as a leak, which should be investigated before starting the measurements. The coefficient of correlation can be calculated with Pearson's equation or extracted from the graphical solution. Examples of the calibration calculations with numeric values can be found in various places in the literature e.g. (Tisch, 2015) or (APTI, 1980).

There are several aspects that can lead to an erroneous calibration, related to operator mistakes and technical issues with the sampler. In both cases, the results obtained from the measurement may be meaningless. One way of identifying a flawed calibration is to operate two Hi-Vol samplers near each other (co-located sampling). This method is similar to analyzing duplicate laboratory samples and is expected to result in similar calibration results. When significant differences between the co-located samplers occur, the calibration procedure and the technical integrity of the samplers should be investigated.

## 4 Conclusion

This paper provides a missing piece of information in the literature regarding air sampling in the environment, showing that by its nature, air sampling is a more complex process than sampling other environmental matrices. We have shown the variables and derivation of the equations used for calculating the air flow rate through a Hi-Vol air sampler, and the process used for calibration of that flow rate. This allows investigators to identify the mass of contaminant found in a volume of air, once the analytical work has been completed. This detailed explanation of the process and equations allows a deeper understanding of the required variables and can be used for error estimation purposes.

**Author contribution**

M.H. devised the idea for the paper, researched the literature, wrote the introduction part, and provided feedback on the manuscript. R.H. developed the theoretical framework, derived the equations, wrote the paper, and prepared the figures.

**Acknowledgments**

This work was originally prepared as part of the air sampling curriculum for the course "AT-331 Arctic Environmental Pollution – Atmospheric Distribution and Processes", a master and PhD-level course offered at the University Center in Svalbard (UNIS) from 2013 – 2017.

## Notations

The following symbols are used in this paper:

$\rho$ = density $\left[\dfrac{\text{kg}}{\text{m}^3}\right]$

$A$ = area [m²]

$a$ = slope

$b$ = intercept

$C$ = concentration $\left[\dfrac{\text{kg}}{\text{m}^3}\right]$

$c$ = constant

$g$ = standard gravity $\left[\dfrac{\text{m}}{\text{s}^2}\right]$

$h$ = height [m]

$m$ = mass [kg]

$n$ = number of observations [−]

$p$ = pressure [Pa]

$R$ = specific gas constant $\left[\dfrac{\text{J}}{\text{kg K}}\right]$

$r$ = Pearsons correlation coefficient [−]

$T$ = temperature [°K]

$t$ = time [s]

$V$ = volume of air [m³]

$\dot{V}$ = volume flow $\left[\dfrac{\text{m}^3}{\text{s}}\right]$

$v$ = velocity $\left[\dfrac{\text{m}}{\text{s}}\right]$

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

**Figures**

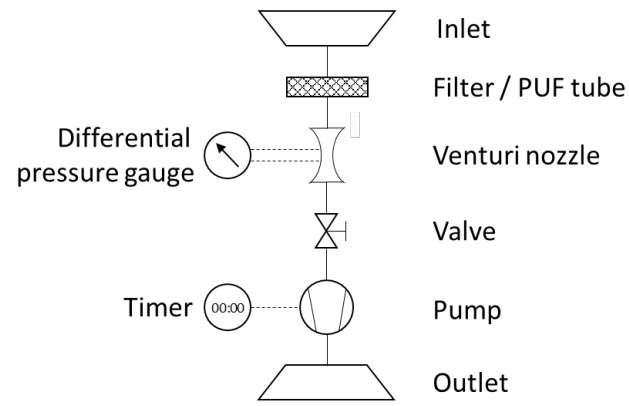

Fig. 1. Main components of a typical high-volume air sampler

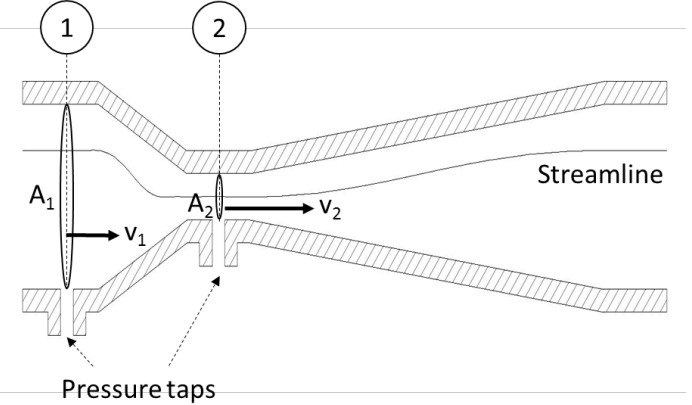

Fig. 2. Venturi nozzle

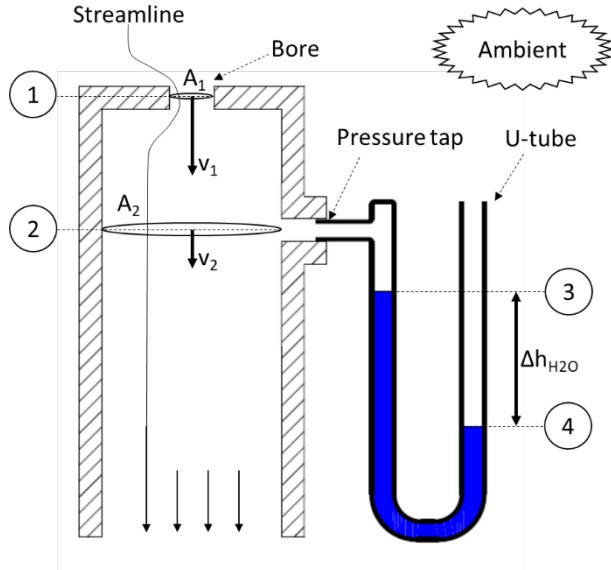

Fig. 3. Orifice plate calibrator

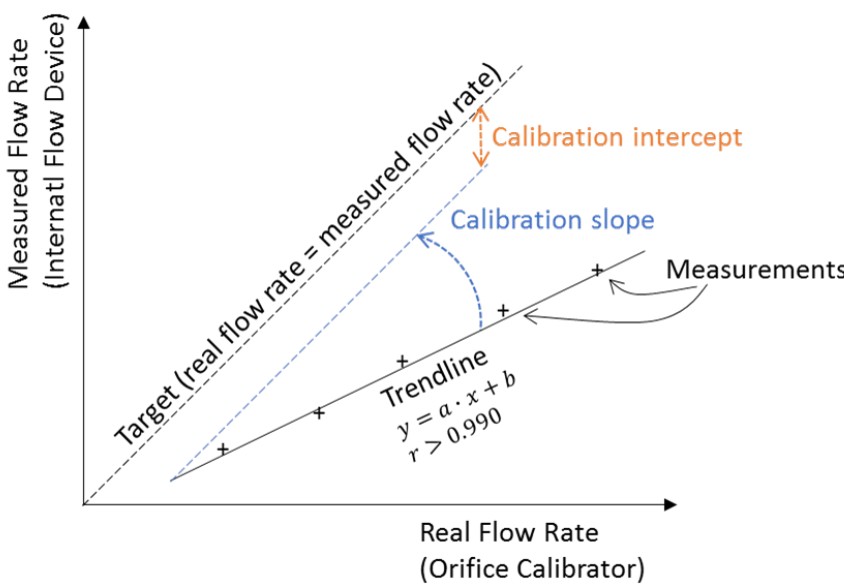

 Fig. 4. Calibration using a linear correlation with intercept and slope