# Peer review of "Derivation of Flow Rate and Calibration Method for High-Volume Air Samplers"

_Atmospheric Measurement Techniques, 2018_

## Short Comment (SC1) · 1 Dec 2018

The main focus of this article is to use the standard flow rate for a hi-vol sampler. Calibration equations are proposed for the orifice type flow meters. However, the correct flow rate to run a pre-separator, PM inlet, is the actual flow rate since the Stokes number depends on flow velocity. In addition, the PM concentration of the aerosols that one breathes is at the actual field conditions and the concentration should also be actual value. Therefore, I disagree with the idea of converting PM concentration into standard concentration based on the standard flow rate which may jeopardize the cut-off characteristic of the pre-separator, also the actual PM that one exposes maybe compromised.

---

## Author Comment (AC1) · 18 Dec 2018

Norwegian University of Science and Technology
* * *
We thank the reviewer for his comment. The question is raised about a Stokes number dependency of the particle matter (PM ) concentration. First of all, we would like to emphasize that we do not propose a new kind of calibration or measurement procedure. High-Volume air sampling is a robust and widely used method (Refs: Jutze & Foster,1967; Clements et al., 1972; Salamova et al. 2014, Salamova et al., 2016) and the main equations for calibration and flow rates are well established (Refs: USEPA, 1999; Tisch, 2015). The point of this paper is to offer a derivation of these established equations and methods that is currently missing in the literature.

Nonetheless, we would like to address the issue. It seems that the reviewer is considering a high-vol sampler with particle pre-separation. Typical flow rates of high-vol samplers such as the e.g. the Tisch TE-1000 are in the range of 200-400 L/min. Typical factors between the standardized (i.e. density adjusted) and actual volume flow rate can be calculated with $\rho/\rho_0$ and range from 0.94 (T=35°C) to 1.16 (T=-25°C). Assuming a case with a flow rate of 300L/min and an inlet section of 10cm², this translate into flow velocities of 0.5m/s and a Reynolds number in the order of $10^0$ which may indicate Stokes flow. In such cases the relaxation time of a 10micron particle with a density of 1000kg/m³ can be estimated to 0.0003s which results in a Stokes number in the order of magnitude $10^{-3}$. For low Stokes numbers it may be assumed that the particles will follow the streamlines. The aforementioned correction factor in the order of 0.94-1.16 is therefore very unlikely to change the particle behaviour in a significant way. Nonetheless, we acknowledge that there is a possibility of a relevant effect for some cases. Hence, we will include a comment in the paper addressing this issue and highlighting that the method explained in the paper can be easily adjusted to represent real flow (instead of density standardized flow) by omitting Equation 10.

Best regards,
Richard Hann

**Richard Hann**, PhD candidate
Phone: +47 48 020 891
Mail: richard.hann@ntnu.no

**Norwegian University of Science and Technology (NTNU)**
Autonomous Marine Operations and Systems (AMOS)
Centre for Integrated Remote Sensing and Forecasting for Arctic Operations (CIRFA)
Department of Engineering Cybernetics (ITK)
Address: O.S. Bragstads plass 2D, NO7491 Trondheim, NORWAY

---

## Short Comment (SC2) · 28 Dec 2018

A PM10 inlet normally has an impactor to remove particles greater than 10 micrometer in aerodynamic diameter. The actual flow velocity thru the nozzles is important to determine the right cutoff diameter. That is, the actual flow rate should be used and controlled properly. Therefore, the reply from the author can't be accepted.
* * *

---

## Author Comment (AC2) · 14 Jan 2019

The comments from this reviewer show a misunderstanding of the nature of sampler that we discuss in our manuscript. We are not deriving the equations for a PM-10 sampler assumed by the commenter; we do not mention PM-10 sampling anywhere in the manuscript. We are deriving equations for Hi-Vol samplers that do not discriminate on the basis of particle size as PM-10 samplers are designed to do.

We have provided references that the commenter can refer to in order to understand the operation of Hi-Vol samplers. One manufacturers operating manual can be found here: https://tisch-env.com/wp-content/uploads/2015/07/TE-1000-PUF-Manual.pdf.

---

## Short Comment (SC3) · 25 Jan 2019

The authors should highlight how the information presented in the paper builds on existing methods papers and coursework such as : https://www.apti-learn.net/lms/register/display_document.aspx?dID=73.
* * *

---

## Referee Comment (RC1) · Ellickson (Referee) · 30 Jan 2019

The authors should highlight how the information presented in the paper builds on existing methods papers and coursework such as : https://www.apti-learn.net/lms/register/display_document.aspx?dID=73.
* * *

---

## Referee Comment (RC2) · Anonymous Referee #2 · 27 Mar 2019

Comments on amt-2018-301, "Derivation of Flow Rate and Calibration Method for High-Volume Air Samplers"

Overall I think this paper does a fine job showing the derivations of the equations necessary for the calibration of a high volume sampler. I think the paper as a few major shortcomings that I outline below.

**Specific Comments**
- Firstly the manuscript fails to fully consider the body of work it is contributing to. While I understand it is impossible to encapsulate the very large body of work about Hi-Vol samples greater effort needs to put forth to support statements made in the introduction.
- Similarly to above, and as mentioned in the interactive discussion section, very similar derivations have been discussed before. While this particular derivations has not be published in the peer-reviewed literature, further acknowledgements should be made to similar derivations.
- This paper has important applications for atmospheric measurements and I think the paper would greatly benefit from a discussion in the conclusions section addressing the implications surrounding an improperly calibrated Hi-Vol and the potential ramifications.
- This paper would greatly benefit from real data showing the application of this approach. I do not know if this data is readily available to the authors, but it would go a long way in illustrating the theory.

**Technical Corrections**
- Page 1 Line 16: The first sentence is phrased weird, suggest saying "collecting atmospheric environmental samples is inherently…"
- Page 2 Line 1: the Hi-Vol abbreviation has not yet been defined.
- Page 2 Line 24: They are usually called polyurethane plugs not tubes.
- Page 3 Line 2: Weird phrasing, suggest "Results may exhibit large variability due to errors in weighing,…"

---

## Author Response (AR1)

**Response to Referee Comments**

Chuen-Jinn Tsa's comment #1:

> The main focus of this article is to use the standard flow rate for a hi-vol sampler. Calibration equations are proposed for the orifice type flow meters. However, the correct flow rate to run a pre-separator, PM inlet, is the actual flow rate since the Stokes number depends on flow velocity. In addition, the PM concentration of the aerosols that one breathes is at the actual field conditions and the concentration should also be actual value. Therefore, I disagree with the idea of converting PM concentration into standard concentration based on the standard flow rate which may jeopardize the cut-off characteristic of the pre-separator, also the actual PM that one exposes maybe compromised.

Author's response:

> We thank the reviewer for his comment. The question is raised about a Stokes number dependency of the particle matter (PM ) concentration. First of all, we would like to emphasize that we do not propose a new kind of calibration or measurement procedure. High-Volume air sampling is a robust and widely used method (Refs: Jutze & Foster,1967; Clements et al., 1972; Salamova et al. 2014, Salamova et al., 2016) and the main equations for calibration and flow rates are well established (Refs: USEPA, 1999; Tisch, 2015). The point of this paper is to offer a derivation of these established equations and methods that is currently missing in the literature.

> Nonetheless, we would like to address the issue. It seems that the reviewer is considering a high-vol sampler with particle pre-separation. Typical flow rates of high-vol samplers such as the e.g. the Tisch TE-1000 are in the range of 200-400 L/min. Typical factors between the standardized (i.e. density adjusted) and actual volume flow rate can be calculated with and range from 0.94 (T=35°C) to 1.16 (T=-25°C). Assuming a case with a flow rate of 300L/min and an inlet section of 10cm², this translate into flow velocities of 0.5m/s and a Reynolds number in the order of which may indicate Stokes flow. In such cases the relaxation time of a 10micron particle with a density of 1000kg/m³ can be estimated to 0.0003s which results in a Stokes number in the order of magnitude . For low Stokes numbers it may be assumed that the particles will follow the streamlines. The aforementioned correction factor in the order of 0.94-1.16 is therefore very unlikely to change the particle behaviour in a significant way. Nonetheless, we acknowledge that there is a possibility of a relevant effect for some cases. Hence, we will include a comment in the paper addressing this issue and highlighting that the method explained in the paper can be easily adjusted to represent real flow (instead of density standardized flow) by omitting Equation 10.

Author's changes:

Added text for clarification that this paper is mainly intended to be used for regular Hi-Vol samplers and that the equations may not be applicable for samplers with particle pre-separators mentioned by the reviewer.

*"Some specialty Hi-Vol samplers, including those for $PM_{10}$, $PM_{2.5}$, and total suspended particulates (TSP) have different flow devices (e.g. particle pre-separators) or different metering systems, so the equations derived and conditions discussed here may not fully apply to them."*
* * *
Chuen-Jinn Tsa's comment #2:

A PM10 inlet normally has an impactor to remove particles greater than 10 micrometer in aerodynamic diameter. The actual flow velocity thru the nozzles is important to determine the right cutoff diameter. That is, the actual flow rate should be used and controlled properly. Therefore, the reply from the author can't be accepted.

Author response:

The main focus of this article is to use the standard flow rate for a hi-vol sampler. Calibration equations are proposed for the orifice type flow meters. However, the correct flow rate to run a pre-separator, PM inlet, is the actual flow rate since the Stokes number depends on flow velocity. In addition, the PM concentration of the aerosols that one breathes is at the actual field conditions and the concentration should also be actual value. Therefore, I disagree with the idea of converting PM concentration into standard concentration based on the standard flow rate which may jeopardize the cut-off characteristic of the pre-separator, also the actual PM that one exposes maybe compromised.

Author's changes:

See above.
* * *
Kristie Ellickson's comment #1 & #2 (identical comments):

The authors should highlight how the information presented in the paper builds on existing methods papers and coursework such as : https://www.aptilearn.net/lms/register/display_document.aspx?dID=73.

Author's response:

This commenter has provided a link to a document of exercises in one of the courses offered by the Air Pollution Training Institute. The document is undated, but related documents from EPA were prepared in the late 1970s. Within the document are equations for calibration of total suspended solids (TSP) air samplers. These samplers use the same calibration equations

as PUF samplers, which are the general topic of our manuscript. As we note in the beginning of our paper, none of the methods or documents we have seen show the derivation of the calibration equations. The same is true with this document, and we will include the reference as further evidence of the need for the derivation. C1 AMTD Interactive comment Printer-friendly version Discussion paper It is important to point out that TSP samplers are operationally different from PUF samplers. One of the significant differences is that on a PUF sampler, what is being calibrated is the differential pressure gauge (e.g. Magnehelic$^{TM}$). A TSP sampler does not have a differential pressure gauge but instead uses a device referred to in these exercises as a "recording transducer" (Laboratory 1, Figure 2) which in other words is chart paper, shown in Figure 7, and problem set 1 Figure 3. That is what is being calibrated. Part of the resulting difference is that if the TSP sampler experiences variability in flow, it will appear on the chart paper. However, TSP samplers are equipped with Mass Flow Controllers (MFC) which sense the rate of air flow and will adjust the flow (by adjusting motor speed) to keep it constant. A PUF sampler does not have a chart recorder or a MFC. The PUF operator accounts for the difference in the flow of the sampler by calculating an average differential pressure gauge values at the beginning and end of the sampling event. In this document here are three experiments, none of which are involved in our paper or, in general, in use of a PUF sampler. Experiment 1 is the calibration of the orifice device using a Roots meter. We mention this in our manuscript beginning on page 6, line 13. For us, this is typically performed by the manufacturer in a laboratory environment. Experiment 2 is Reference flow audit device which calibrates the resistance plates. We do not use resistance plates to change flow during calibration. We use the ball valve on the sampler to change the flow. Experiment 3 is verification of mass flow controller (MFC) range. Our devices are not TSP samplers and do not use an MFC.

Further response to the APTI course document, 2019-1-30 On page 2 of our manuscript, beginning on line 5, we have a paragraph that specifically notes the many available documents that offer no information about the derivation of the equations that are used for air Hi-Vol sampler calibration. The link provided by this commenter is to a series of exercises relevant to a TSP sampler, not a PUF sampler. In any case, some of the calibration equations presented in the TSP exercise are the same or equivalent as used for the PUF sampler. We will use APTI as further evidence of missing derivation of calibration equations and explain the similarities between equations.

Author's changes:

Following the reviewers recommendation, we have included the ATPI coursework as another example for the literature that does not derive the equations used for the calibration of Hi-Vol samplers.

Anonymous Referee #2's comment #1:

Overall I think this paper does a fine job showing the derivations of the equations necessary for the calibration of a high volume sampler. I think the paper as a few major shortcomings that I outline below.

**Specific Comments**

• Firstly the manuscript fails to fully consider the body of work it is contributing to. While I understand it is impossible to encapsulate the very large body of work about Hi-Vol samples greater effort needs to put forth to support statements made in the introduction.

• Similarly to above, and as mentioned in the interactive discussion section, very similar derivations have been discussed before. While this particular derivations has not be published in the peer-reviewed literature, further acknowledgements should be made to similar derivations.

• This paper has important applications for atmospheric measurements and I think the paper would greatly benefit from a discussion in the conclusions section addressing the implications surrounding an improperly calibrated Hi-Vol and the potential ramifications.

• This paper would greatly benefit from real data showing the application of this approach. I do not know if this data is readily available to the authors, but it would go a long way in illustrating the theory.

**Technical Corrections**

• Page 1 Line 16: The first sentence is phrased weird, suggest saying "collecting atmospheric environmental samples is inherently…"

• Page 2 Line 1: the Hi-Vol abbreviation has not yet been defined. • Page 2 Line 24: They are usually called polyurethane plugs not tubes.

• Page 3 Line 2: Weird phrasing, suggest "Results may exhibit large variability due to errors in weighing,…"

Author's response:

We thank the reviewer for both his specific comments and the technical correction. Below we are address his comments in detail.

 **Specific Comments**

- We have carefully reviewed the literature and included the most relevant papers so far. While there is a large amount of papers regarding the use of high-volume samplers, only a very limited number of papers concern the actual measurement and calibration procedure. We are open to include more references, but unfortunately, the reviewer does not recommend any specific papers.
- Nonetheless, we revisited the literature and found four additional sources that support the purpose of our paper. All the following papers include some discussion about the calibration procedure, however none explains the used equations in the

same level of detail as we do. The following references will be included into the final manuscript to account for previous work as well as similar derivations:

- o Lee Jr, R. E., Caldwell, J. S., & Morgan, G. B. (1972). The evaluation of methods for measuring suspended particulates in air. Atmospheric Environment (1967), 6(9), 593-622.
- o Lynam, D. R., J. O. Pierce, and J. Cholak. "Calibration of the High-Volume Air Sampler." American Industrial Hygiene Association Journal 30.1 (1969): 83-88.
- o ATPI 435 Atmospheric Sampling Course
- o US EPA Appendix B of 40 CFR 50

- We agree that including a discussion regarding the effect of improperly calibrated Hi-Vol samplers would add relevance to the paper. This will be included in the finalized version.

- We believe including data is a good idea but gets away from the motivation of this paper to highlight the physics behind the calibration process. We feel that the pure calculation part is well covered in the literature – it is the background on where the equations come from that is missing. However, we will include a remark where such an example can be found (i.e. in the Tisch Environmental manuals).

**Technical Corrections**

- Will be implemented as suggested.

Author's changes:

- Additional references (Lee et al. 1972), (Lynam et al., 1969),(APTI, 1980) and 40 CFR 50, AppB, have been added
- Added note on where to find examples for the calibration:
  "Examples of the calibration calculations with numeric values can be found in various places in the literature e.g. (Tisch, 2015) or (APTI, 2012)."
- Added text discussing erroneous calibration:
  "There are several aspects that can lead to an erroneous calibration, related to operator mistakes and technical issues with the sampler. In both cases, the results obtained from the measurement may be meaningless. One way of identifying a flawed calibration is to operate two Hi-Vol samplers near each other (co-located sampling). This method is similar to analyzing duplicate labratory samples and is expected to result in similar calibration results. When significant differences between the co-located samplers occur, the calibration procedure and the technical integrity of the sampler should be investigated."

**Compare Results**

| Old File: | | New File: |
|---|---|---|
| **amt-2018-301-manuscript-version2 (1).pdf** | versus | **Hann & Hermanson AMT 2018_v8.pdf** |
| **10 pages (314 KB)** | | **12 pages (528 KB)** |
| 06/09/2018 14:22:50 | | 27/06/2019 07:18:08 |

**Total Changes**

**650**

**Content**

90    Replacements

79    Insertions

65    Deletions

**Styling and Annotations**

416   Styling

0     Annotations

Go to First Change (page 1)

[revised manuscript text omitted]